

# Breathlessness and "exacerbation" questions predictive for incident COPD (MARKO study): data after two years of follow-up

Žarko Vrbica[1,2,*], Justinija Steiner[3,*], Marina Labor[4], Ivan Gudelj[5] and Davor Plavec[6,7]

[1] Medical Nursing, University of Dubrovnik, Dubrovnik, Croatia, Dubrovnik, Croatia
[2] Pulmonology and Immunology, Dubrovnik General Hospital, Dubrovnik, Croatia, Croatia
[3] Osijek-Baranja Country Medical Center, Osijek, Croatia, Osijek, Croatia
[4] Cancer and Lung Health Care Unit, University Hospital at Linköping, Linköping, Sweden
[5] Medical Faculty, University of Split, Split, Croatia
[6] Research Department, Prima Nova, Zagreb, Croatia
[7] Medical Faculty, Josip Juraj Strossmayer University of Osijek, Osijek, Croatia
* These authors contributed equally to this work.

Corresponding author
Davor Plavec,
davorplavec@gmail.com

## ABSTRACT

**Aims:** To determine the predictability of the MARKO questionnaire and/or its domains, individually or in combination with other markers and characteristics (age, gender, smoking history, lung function, 6-min walk test (6 MWT), exhaled breath temperature (EBT), and hsCRP for the incident chronic obstructive pulmonary disease (COPD) in subjects at risk over 2 years follow-up period).

**Participants and Methods:** Patients, smokers/ex-smokers with >20 pack-years, aged 40–65 years of both sexes were recruited and followed for 2 years. After recruitment and signing the informed consent at the GP, a detailed diagnostic workout was done by the pulmonologist; they completed three self-assessment questionnaires—MARKO, SGRQ and CAT, detailed history and physical, laboratory (CBC, hsCRP), lung function tests with bronchodilator and EBT. At the 2 year follow-up visit they performed: the same three self-assessment questionnaires, history and physical, lung function tests and EBT.

**Results:** A sample of 320 subjects (41.9% male), mean (SD) age 51.9 (7.4) years with 36.4 (17.4) pack-years of smoking was reassessed after 2.1 years. Exploratory factor analysis of MARKO questionnaire isolated three distinct domains (breathlessness and fatigue, "exacerbations", cough and expectorations). We have determined a rate for incident COPD that was 4.911/100 person-years (95% CI [3.436–6.816]).
We found out that questions about breathlessness and "exacerbations", and male sex were predictive of incident COPD after two years follow-up (AUC 0.79, 95% CI [0.74–0.84], $p < 0.001$). When only active smokers were analyzed a change in EBT after a cigarette ($\Delta$EBT) was added to a previous model (AUC 0.83, 95% CI [0.78–0.88], $p < 0.001$).

**Conclusion:** Our preliminary data shows that the MARKO questionnaire combined with EBT (change after a cigarette smoke) could potentially serve as early markers of future COPD in smokers.

## INTRODUCTION

Chronic obstructive pulmonary disease (COPD) is one of the top three causes of death worldwide and represents an important public health challenge that is both preventable and treatable (*Mathers & Loncar, 2006*; *GOLD, 2023*). COPD is characterized by persistent airflow limitation that is typically progressive and associated with an enhanced chronic inflammatory response in the airways and lung tissue to harmful particles or gases (*Vestbo et al., 2013*).

Early detection and treatment of COPD can help manage symptoms and slow the progression of the disease. Because of that, GOLD advocates active case finding in patients with symptoms and/or risk factors, but not screening spirometry. Wide reference values for spirometry prevent earlier diagnosis and spirometry from being used for screening, and simple portable devices are not widely accepted for COPD case finding (*Labor et al., 2016b*). Finding patients with early COPD is already difficult, but even those diagnosed in an early stage of disease (based on spirometry criteria) have already an advanced disease in the biological point of view. Clinically, early COPD is in patients represented with a developed airflow limitation due to the widespread damage to the bronchi and lung parenchyma. To be able to prevent these pathohistological damage, we should detect the disease in the biologically early stage before the irreversible changes occurred (*Martinez et al., 2022*).

Cigarette smoking is a key environmental risk factor for COPD, but fewer than 50% of heavy smokers develop COPD (*Lundbäck et al., 2003*). Since not all smokers and not even all the patients with pre-COPD (individuals with structural lung lesions and/or physiological abnormalities but without airflow obstruction) and Preserved Ratio Impaired Spirometry (PRISm) will eventually develop overt COPD, it is important to find the predictive markers that can detect the patients with early pathophysiological changes and find the way to protect them from further progression to clinically manifest COPD.

As the measurement of the exhaled breath temperature (EBT) using a portable device was found easy, non-invasive and reliable for monitoring airway inflammation it was used in smokers and COPD for research (*Popov et al., 2017*). It was shown by a group of authors that EBT increases in acute exacerbation of COPD and may be related to airway inflammation (*Lázár et al., 2014*). Smoking has been found to cause an acute increase in airways inflammation but not in all smokers. Change in EBT after smoking a cigarette in patients without a diagnosis of COPD and in GOLD 1 stage at initial assessment was significantly predictive for disease progression after 2 years (*Labor et al., 2016a*). Thus it was shown that EBT could have a potential to predict the future development of COPD at least in a proportion of subjects (current smokers). This data still needs to be confirmed by future studies.

So one of the aims of the MARKO project (Early detection of COPD patients in GOLD 0 (smokers) population–MARKO project) (*Vrbica et al., 2017*) and this analysis was to

assess the predictability of MARKO questionnaire for incident COPD during the 2-years follow-up.

## MATERIALS AND METHODS

The aim, design and setting of the study: This analysis is the part of the broader research project "Early Detection of COPD Patients in GOLD 0 (Smokers) Population–MARKO Project", a two-phase prospective observational cohort study in subjects at risk for COPD to identify individuals that will further on develop COPD. The MARKO project is a prospective, observational, non-interventional cohort study of subjects at risk for the development of COPD. The study was in detail described previously (*Vrbica et al., 2017*) and registered at https://clinicaltrials.gov/ct2/show/NCT01550679. The study was approved by the local Ethics Committee (CHS, 02/2009) and conducted according to the recent version of the Declaration of Helsinki, Good Clinical Practice (GCP), and other relevant international and national laws. All participants signed the informed consent form (ICF) before starting any procedure related to the study (*Vrbica et al., 2017*).

In brief the project was organized in two phases (Fig. 1). Phase I recruited 450 subjects with the risk of COPD by 25 GPs in and around four major cities, during any (unrelated to respiratory problems) visit, if they satisfied inclusion/exclusion criteria and signed ICF. Inclusion criteria were: smokers/ex-smokers; both sexes; aged 40–65 years at inclusion; smoking history ≥20 pack-years (calculated as a number of cigarettes smoked per day multiplied by the number of years of smoking divided by 20); and no previous diagnosis of COPD. Exclusion criteria were: any clinically relevant chronic disease significantly affecting HRQoL; immunosuppressive therapy; acute respiratory disease 4 weeks before inclusion; hospitalization during the past 3 months; myocardial infarction (MI), cerebrovascular infarction (CVI) or transient ischemic attack (TIA) during the past 6 months; diagnosis of asthma; and an inability to perform the diagnostic protocol. Subjects filled out the MARKO questionnaire and measured lung function at GP's office and were referred to pulmonologist. There a diagnostic workup consisting of the MARKO questionnaire, HRQoL questionnaires (SGRQ and CAT), history and physical, exhaled breath temperature (EBT) before (EBTb) and after a smoked cigarette (EBTc), lung function testing with bronchodilator (salbutamol), lung diffusion capacity (DLCO), blood sampling (hematology; highly sensitive C-reactive protein (hs-CRP)), 6-min walk test (6MWT), and the assessment for diagnosis and severity of COPD were done.

Phase II included subjects from phase I assessed as 'healthy' smokers, symptomatic smokers (GOLD 0) or as COPD GOLD 1 that were followed and reassessed after 2 years (±2 months) after baseline by the same pulmonologist. Incident COPD diagnosis was made by the same trained pulmonologist in a tertiary level hospital based on clinical presentation and lung function according to GOLD. Incidence of newly diagnosed COPD after 2-years of follow up was used to identify diagnostic parameters that are most sensitive for early impairment in COPD, to determine the predictability of developed screening MARKO questionnaire alone or with other markers of early impairment in COPD. The flow-chart of the MARKO study is presented in Fig. 1.

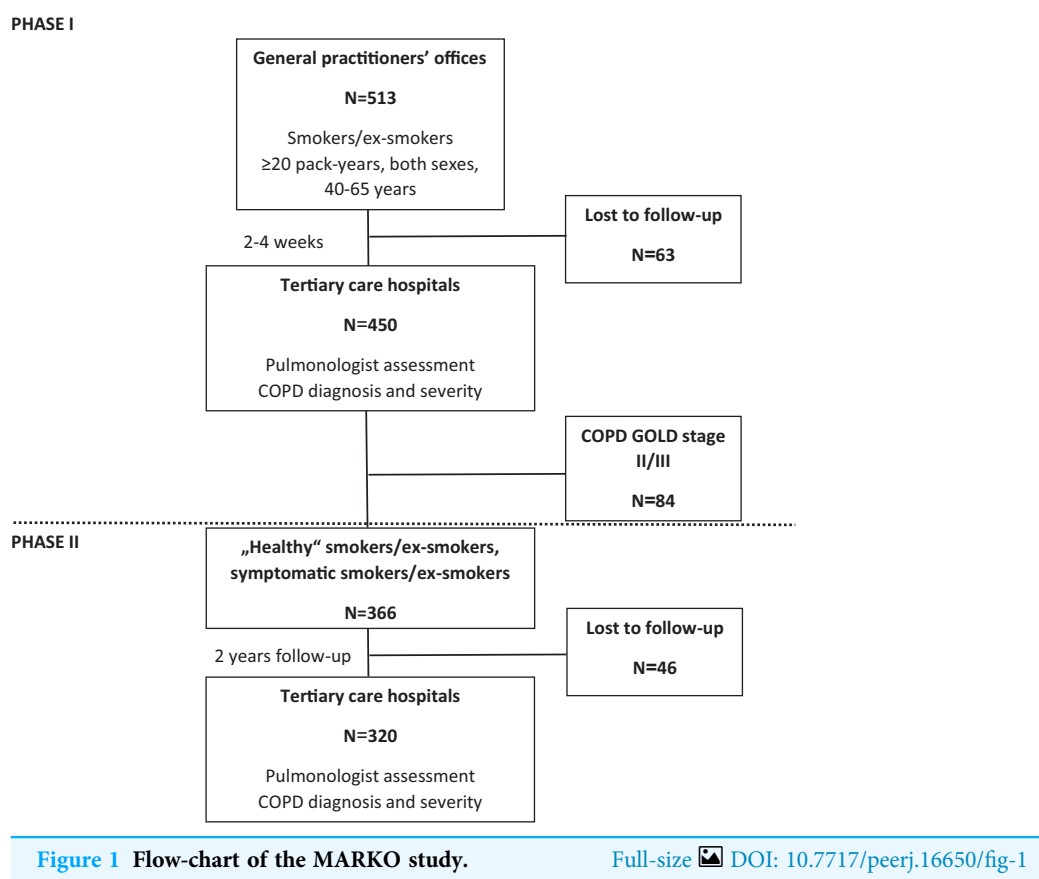

**Figure 1 Flow-chart of the MARKO study.**

This research attempts to answer two following questions: (1) can we identify cheap and simple tools that will allow a precise and accurate identification of future incident COPD from a population at risk; (2) is there a combination (pattern) of tools, functional parameters, genetic and biochemical markers that can reliably predict incident COPD in a population at risk, thus allowing early intervention.

Primary end point of this analysis was to assess the predictability of newly constructed self-administered health related quality of life (HRQoL) questionnaire (MARKO questionnaire) and its domains to be used alone or in combination with other markers (exhaled breath temperature (EBT), lung function, inflammatory markers) in identification of subjects who will develop COPD during the 2-year follow-up.

Secondary endpoint: to determine the rate of progression of COPD in patients with GOLD 0 during the follow-up.

## Methods

**Screening questionnaire (MARKO questionnaire).** The MARKO questionnaire is a newly constructed HRQoL questionnaire developed by a group of experts. The questionnaire and the psychometric characteristics were already published elsewhere (*Vrbica et al., 2016*). The questionnaire comprises 18 questions covering the manifestation and frequency of the symptoms already present at early stages of COPD (Supplemental Material). The total scores ranged 0 to 57 with higher scores indicating poorer HRQoL.

**St. George Respiratory Questionnaire (SGRQ).** The SGRQ is a standardized self-administered airways disease-specific questionnaire divided into three subscales: symptoms (eight items), activity (16 items), and impacts (26 items). SGRQ scores were calculated using the Excel® SGRQ calculator with scores ranging from 0 (no impairment) to 100 (maximum impairment) (*Jones et al., 1992*).

**COPD Assessment Test (CAT).** The CAT is a validated, short (eight-item) patient completed questionnaire developed for use in routine clinical practice to measure the health status of patients with COPD, with scores ranging from 0 (no impairment) to 40 (maximum impairment) (*Jones et al., 2009*).

**Lung function.** Spirometry was performed using computerized pneumotachographs (Jaeger®, CareFusion, San Diego, CA, USA) using the same procedure at all clinical sites (lung function labs at tertiary hospitals) in agreement with the ATS/ERS standardization (*Quanjer et al., 2012*). Bronchodilator test was done with repeated spirometry 20 min after inhalation of 400 mcg of salbutamol using the inhalation chamber. Spirometric parameters (FVC, FEV1, FEV1/FVC ratio, peak expiratory flow (PEF), forced expiratory flow between 25 and 75% FVC (FEF25-75)) were recorded as absolute values and as percentage of predicted according to GLI (*Quanjer et al., 2012*). Single-breath diffusing capacity of the lung for carbon monoxide (DLCO) was measured using a rapid carbon monoxide and helium analyzer (Ganzhorn, Germany), which was calibrated prior to each measurement. Values for DLCO and DLCO corrected for alveolar volume (VA) (DLCO/VA) were obtained and are reported as percent predicted values (*Roca et al., 1990*).

**Exhaled breath temperature (EBT).** EBT was measured using X-Halo® device (Delmedica Investments, Singapore) according to a previously validated method (*Popov et al., 2007*). EBT was measured twice on the same occasion (during the initial visit); (1) baseline measurement before any other procedure (lung function, bronchodilator test, 6 MWT) at least 1h after the last smoked cigarette (EBTb) and (2) done only in active smokers 15 min after a smoked cigarette (EBTc) and recorded with precision of 1/100 of a °C. No other procedure apart from cigarette smoking was carried out between the two EBT measurements.

**Blood sampling.** Blood samples for serum were drawn at minimal volume of 3 ml in serum separation vacuum tubes (containing Z Serum Clot Activator gel) and sent to the laboratory. Samples were kept at room temperature for at least 30 min, but had to be processed within 2 h of blood collection. Upon centrifugation at 3,000 rpm for 10 min, a minimum of 300 µl of serum was separated for further detection of hs-CRP level. Blood samples for hematology were drawn at minimal volume of 3 ml in K2EDTA vacuum blood collection tubes and sent to the laboratory for complete blood cell hematological analysis. Laboratory analyses were done in local laboratories and included complete blood count, white blood cells differential count, hematocrit, hemoglobin and hs-CRP.

Functional exercise capacity was assessed using 6 MWT according to the ATS guidelines and expressed as walked distance in meters and as % of predicted according to *Troosters, Gosselink & Decramer (1999)*.

## Data analyses

Data analyses were done using STATISTICA version 12 (StatSoft, Inc., Tulsa, OK, USA), and MedCalc Statistical Software version 22 (MedCalc Software bvba, Ostend, Belgium; https://www.medcalc.org; 2023). Sample size calculation was done based on the following assumptions: we expected to find 25% patients in different stages of COPD according to the airflow limitation. We therefore expected that the sample would consist of 75% of 'healthy' and symptomatic smokers, 12.5% patients with COPD GOLD 1, 6.25% in GOLD 2 and 6.25% in GOLD 3 or 4 stages at initial visit with the expected difference in the MARKO questionnaire scores of two points and SD of 2.5 points between 'healthy' smokers *vs* symptomatic smokers *vs*. COPD GOLD 1/2 having a statistical power of >80% with alpha = 0.05 for the sample size of at least 500 subjects. The expected yearly incidence rate was planned at 10/per 100 patient-years.

Categorical data were presented as absolute numbers and percentages. Quantitative data were expressed as mean and standard deviation (SD) or median and interquartile range (IQR) with regards to the type of distribution. Normality of distribution was assessed using Kolmogorov-Smirnov test. Categorical data were compared between subgroups using chi-square test or Fisher exact test and continuous variables using Mann-Whitney U-test and Kruskal-Wallis ANOVA (non-normal distribution expected). Secondary validation of the MARKO questionnaire was done to determine the construct validity and predictability (alone or in combination with other markers) for future development and/or progression of COPD. Construct validity of the MARKO questionnaire was assessed using factorial analysis to confirm the number of factors with calculations of inter- and intracorrelations between the factors and items. Utility of different markers for disease progression was assessed using logistic regression models and expressed as odds ratio (OR) with 95% confidence intervals (CIs). Predictive power for the models was presented using receiver operator curve (ROC) analysis with AUC (with 95% CIs). $p < 0.05$ was used as statistically significant for all analyses with the correction for multiple comparisons.

## RESULTS

Three hundred and twenty patients (186 women) aged 40–65 years at entry were included into this cohort study. Women had a mean (SD) age of 51.4 (7.1) years, and men 52.5 (7.8) years (t = 1.292, $p = 0.198$) with a mean smoking history of 31.7 (14.0) years and 42.9 (19.5) pack-years, men having a significantly greater cumulative exposure (t = 5.677, $p < 0.001$). BMI was 26.12 (4.51) kgm-2 in women and 27.42 (3.90) kgm-2 in men (t = 2.753, $p = 0.006$). Post BD FEV1/FVC was significantly greater in women than in men (mean ± SD; 0.79 ± 0.06 *vs* 0.77 ± 0.07, t = 3.087; $p = 0.002$), with comparative post BD FVC (98.0 ± 12.6% *vs* 97.3 ± 13.3%, t = 0.475; $p = 0.635$) and FEV1 (97.2 ± 14.4% *vs* 95.3 ± 15.1%, t = 1.132; $p = 0.259$). 6MWT (expressed as the % of predicted) was significantly lower in men (mean ± SD; 64.4 ± 10.8% *vs* 60.4 ± 13.3%, t = 2.867; $p = 0.005$).

In Table 1 results of the exploratory factor analysis are presented as factor loadings. Three distinct domains were found with first domain representing questions about different levels of breathlessness and fatigue, second domain representing questions about cough and phlegm and third domain representing questions about "exacerbations" (severe

Table 1 Factor analysis for MARKO questionnaire done at baseline (N = 320).

| Variable | Factor loadings | | |
|---|---|---|---|
| | Factor (1) | Factor (2) | Factor (3) |
| MQq 1 | 0.121 | **0.865** | 0.137 |
| MQq 2 | 0.114 | **0.850** | 0.083 |
| MQq 3 | **0.573** | 0.444 | 0.200 |
| MQq 4 | 0.069 | 0.184 | **0.860** |
| MQq 5 | 0.128 | 0.037 | **0.878** |
| MQq 6 | **0.688** | 0.343 | 0.172 |
| MQq 7 | **0.636** | 0.025 | 0.360 |
| MQq 8 | **0.712** | 0.065 | 0.211 |
| MQq 9 | **0.771** | 0.033 | 0.155 |
| MQq 10 | **0.726** | 0.302 | 0.027 |
| MQq 11 | **0.687** | 0.412 | 0.001 |
| MQq 12 | **0.663** | 0.373 | −0.075 |
| MQq 13 | **0.798** | 0.128 | 0.119 |
| MQq 14 | **0.766** | 0.029 | 0.065 |
| MQq 15 | **0.807** | 0.129 | 0.024 |
| MQq 16 | **0.802** | 0.048 | 0.040 |
| MQq 17 | **0.602** | 0.366 | 0.021 |
| MQq 18 | **0.636** | 0.390 | 0.016 |
| Expl.Var | 7.081 | 2.548 | 1.831 |
| Prp.Totl | 0.393 | 0.142 | 0.102 |

Note:
MQq, MARKO questionnaire question number; bold text represents significant factor loadings.

cold with cough or bronchitis and use of antibiotics for a severe cold with cough or bronchitis) during the past year.

## Follow-up visit

Changes between baseline and follow-up visit are presented in Table 2. Not many changes were seen when the sample was assessed as a whole group. It can be seen that as expected smoking history was significantly greater ($p < 0.001$) after 2 years. Dyspnea (mMRC) was however significantly lower ($p < 0.001$) despite the fact that lung function was significantly worse ($p < 0.001$ for FVC, FEV1, and Tiff), except for post BD PEF ($p = 0.209$). The results of the MQ were significantly better ($p = 0.001$), but CAT score was comparable ($p = 0.238$). SGRQ activity score was worse ($p = 0.043$) and impact, symptoms and total scores were comparable ($p > 0.05$ for all).

## Incident COPD

Incident COPD at the follow-up visit was determined in 33 subjects, with a rate of 4.911/100 patient-years (95% CI [3.436–6.816]). In Table 3 comparison of baseline characteristics between incident COPD group and other subjects is shown. There were significantly more males (75.8% *vs* 37.3%, $p < 0.001$) with a significantly higher smoking

**Table 2  Comparison of baseline and follow-up visit data (N = 320).**

| Variables | Baseline | | Follow-up | | Paired differences | | | |
|---|---|---|---|---|---|---|---|---|
| | Mean | SD | Mean | SD | Mean | SD | 95% CI | $p^a$ |
| Body weight (kg) | 79.04 | 16.11 | 79.05 | 16.73 | 0.01 | 4.93 | [−0.54 to 0.55] | 0.251 |
| Smoking history (p/y) | 36.32 | 17.45 | 38.12 | 18.05 | 1.81 | 2.11 | [2.04–1.57] | <0.001 |
| Comorbidities (No) | 0.68 | 0.79 | 0.63 | 0.80 | −0.05 | 0.80 | [−0.14 to 0.04] | 0.284 |
| Chronic therapy (No) | 0.84 | 1.14 | 0.77 | 1.25 | −0.08 | 1.17 | [−0.21 to 0.05] | 0.144 |
| mMRC | 0.67 | 0.78 | 0.47 | 0.66 | −0.20 | 0.79 | [−0.29 to −0.11] | <0.001 |
| Post BD FVC (L) | 4.10 | 1.03 | 3.84 | 1.00 | −0.27 | 0.44 | [−0.31 to −0.22] | <0.001 |
| Post BD FVC (% predicted) | 97.71 | 12.94 | 92.85 | 13.88 | −4.87 | 10.42 | [−6.04 to −3.70] | <0.001 |
| Post BD FVC (z-score) | −0.17 | 0.94 | −0.52 | 1.00 | −0.34 | 0.76 | [−0.43 to −0.26] | <0.001 |
| Post BD FEV1 (L) | 3.19 | 0.80 | 3.06 | 0.78 | −0.14 | 0.33 | [−0.18 to −0.10] | <0.001 |
| Post BD FEV1 (% predicted) | 96.44 | 14.73 | 94.18 | 15.57 | −2.25 | 9.72 | [−3.34 to −1.16] | <0.001 |
| Post BD FEV1 (z-score) | −0.25 | 1.08 | −0.40 | 1.12 | −0.15 | 0.73 | [−0.23 to −0.07] | <0.001 |
| Post BD TIFF (%) | 78.22 | 6.76 | 80.14 | 7.60 | 1.92 | 5.55 | [1.30–2.53] | <0.001 |
| Post BD TIFF (% predicted) | 98.29 | 8.27 | 101.10 | 9.37 | 2.81 | 7.01 | [2.03–3.58] | <0.001 |
| Post BD MEF25 (L/s) | 1.17 | 0.58 | 1.66 | 5.86 | 0.49 | 5.87 | [−0.17 to 1.16] | 0.010 |
| Post BD MEF50 (L/s) | 3.87 | 1.48 | 3.75 | 1.49 | −0.12 | 0.86 | [−0.21 to −0.02] | 0.002 |
| Post BD PEF (L/s) | 7.70 | 2.09 | 7.59 | 2.17 | −0.11 | 1.40 | [−0.27 to 0.05] | 0.209 |
| MARKO questionnaire (score) | 13.10 | 9.08 | 11.97 | 8.42 | −1.13 | 7.31 | [−1.95 to −0.32] | 0.001 |
| CAT score | 9.73 | 6.98 | 9.15 | 6.81 | −0.58 | 7.05 | [−1.37 to 0.21] | 0.238 |
| SGRQ activity score | 23.72 | 19.82 | 22.59 | 18.28 | −1.13 | 22.76 | [−3.74 to 1.49] | 0.583 |
| SGRQ impact score | 8.06 | 10.75 | 9.17 | 9.71 | 1.11 | 12.93 | [−0.38 to 2.59] | 0.043 |
| SGRQ systems score | 20.39 | 19.19 | 20.42 | 18.31 | 0.03 | 23.84 | [−2.70 to 2.76] | 0.756 |
| SGRQ total score | 14.85 | 12.68 | 15.20 | 11.51 | 0.35 | 14.72 | [−1.35 to 2.05] | 0.342 |

Notes:

[a] Wilcoxon test (paired samples).

p/y , pack-years of smoking; mMRC, modified Medical Research Council dyspnea scale; BD, bronchodilator; FVC, forced vital capacity; FEV1, forced expiratory volume in 1 s; TIFF, Tiffeneau index (FEV1/FVC); MEF25, mid-expiratory flow at 75% of FVC; MEF50, mid-expiratory flow at 50% of FVC; PEF, peak expiratory flow; CAT, COPD Assessment Test; SGRQ, St. George's respiratory questionnaire.

exposure (42.99 *vs* 35.52 pack-years, $p = 0.008$). As more males were present height and weight were also significantly larger ($p < 0.001$ and $p = 0.046$). Incident COPD subjects had significantly more obstruction at baseline with lower post BD values for FEV1, Tiffeneau index and MEF25 and MEF50 ($p < 0.001$ for all). Although DLCO (% predicted) was comparable, KCO (% predicted) was significantly lower ($p = 0.049$). 6MWT showed comparative results, together with dyspnea (mMRC score), EBT, hematology and hsCRP. There were also no difference for CAT score, SGRQ domains and total score and MARKO questionnaire total score. Significant differences were found for answers to question number 4 ($p = 0.014$) of MQ and marginal for questions 3 and 5 ($p = 0.064$ and $p = 0.050$).

Multivariate logistic regression was used to define the baseline variables that can predict incident COPD. The results were shown in Figs. 2 and 3. We found out that questions about breathlessness ("Have you experienced being breathless during preceding 3 months?") and "exacerbations" ("Have you had a severe cold with cough or bronchitis during preceding 12 months?"), and male sex were predictive of incident COPD after

**Table 3 Comparison of baseline measures between the groups of incident COPD and rest (at follow-up visit) (N = 320).**

| Variable | No COPD (n = 287) | | Incident COPD (n = 33) | | Difference | 95% CI | p[a] |
|---|---|---|---|---|---|---|---|
| | Mean | SD | Mean | SD | | | |
| Age (yrs) | 51.76 | 7.38 | 52.38 | 7.80 | 0.62 | [−2.06 to 3.31] | 0.561 |
| Sex (male) | 106 | 37.3% | 25 | 75.8% | | | <0.001 |
| Smoking history (p/y) | 35.52 | 17.01 | 42.99 | 19.18 | 7.47 | [1.23–13.71] | 0.008 |
| Time from baseline (yrs) | 2.11 | 0.21 | 2.14 | 0.22 | 0.03 | [−0.05 to 0.11] | 0.483 |
| Body height (cm) | 171.08 | 9.25 | 177.76 | 9.46 | 6.68 | [3.32–10.04] | <0.001 |
| Body weight (kg) | 78.45 | 15.92 | 84.24 | 17.37 | 5.79 | [−0.03 to 11.61] | 0.046 |
| BMI (kgm$^{-2}$) | 26.69 | 4.32 | 26.52 | 4.45 | −0.17 | [−1.74 to 1.40] | 0.841 |
| Heart rate (min$^{-1}$) | 77.76 | 12.26 | 76.79 | 13.22 | −0.97 | [−5.80 to 3.85] | 0.788 |
| Systolic blood pressure (mmHg) | 127.26 | 14.50 | 131.43 | 12.39 | 4.17 | [−1.42 to 9.76] | 0.123 |
| Diastolic blood pressure (mmHg) | 80.38 | 9.07 | 80.57 | 9.08 | 0.19 | [−3.35 to 3.73] | 0.893 |
| Commorbidities (No) | 0.69 | 0.78 | 0.64 | 0.86 | −0.05 | [−0.34 to 0.24] | 0.599 |
| Chronic treatments (No) | 0.86 | 1.13 | 0.82 | 1.26 | −0.04 | [−0.45 to 0.38] | 0.591 |
| Post BD FVC (L) | 4.04 | 1.00 | 4.66 | 1.15 | 0.62 | [0.25–0.99] | 0.003 |
| Post BD FVC (% predicted) | 97.74 | 12.83 | 97.47 | 14.01 | −0.26 | [−5.03 to 4.50] | 0.760 |
| Post BD FVC (z score) | −0.17 | 0.92 | −0.19 | 1.05 | −0.02 | [−0.36 to 0.33] | 0.749 |
| Post BD FEV1 (L) | 3.20 | 0.80 | 3.13 | 0.85 | −0.07 | [−0.37 to 0.22] | 0.795 |
| Post BD FEV1 (% predicted) | 97.95 | 13.94 | 83.35 | 15.11 | −14.60 | [−19.77 to −9.43] | <0.001 |
| Post BD FEV1 (z score) | −0.14 | 1.02 | −1.21 | 1.12 | −1.07 | [−1.45 to −0.69] | <0.001 |
| Post BD TIFF (%) | 79.51 | 5.62 | 67.21 | 5.55 | −12.30 | [−14.33 to −10.27] | <0.001 |
| Post BD TIFF (% predicted) | 99.81 | 6.89 | 85.24 | 7.60 | −14.58 | [−17.10 to −12.05] | <0.001 |
| Post BD TIFF (z score) | 0.01 | 0.86 | −1.67 | 0.83 | −1.68 | [−1.99 to −1.37] | <0.001 |
| Post BD MEF25 (L/s) | 1.21 | 0.58 | 0.79 | 0.35 | −0.42 | [−0.63 to −0.20] | <0.001 |
| Post BD MEF50 (L/s) | 4.02 | 1.43 | 2.53 | 1.22 | −1.50 | [−2.02 to −0.97] | <0.001 |
| Post BD PEF (L/s) | 7.66 | 2.08 | 8.01 | 2.18 | 0.35 | [−0.43 to 1.13] | 0.199 |
| DLCO (% predicted) | 78.16 | 18.26 | 75.53 | 33.23 | −2.63 | [−10.76 to 5.50] | 0.083 |
| KCO (% predicted) | 79.30 | 19.39 | 71.31 | 23.64 | −7.99 | [−16.02 to 0.04] | 0.049 |
| 6 MWT (m) | 442.39 | 88.85 | 433.81 | 100.29 | −8.57 | [−44.38 to 27.24] | 0.763 |
| 6 MWT (%) | 63.28 | 11.77 | 58.25 | 13.59 | −5.03 | [−9.79 to −0.28] | 0.056 |
| EBTb (°C) | 33.01 | 2.83 | 32.43 | 3.37 | −0.58 | [−1.70 to 0.53] | 0.404 |
| EBTd (°C) | −0.06 | 1.46 | 0.40 | 1.92 | 0.46 | [−0.16 to 1.08] | 0.521 |
| RBC | 4.69 | 0.42 | 4.71 | 0.41 | 0.03 | [−0.14 to 0.20] | 0.626 |
| Hgb | 142.27 | 13.12 | 145.96 | 12.98 | 3.69 | [−1.73 to 9.12] | 0.208 |
| htc | 0.43 | 0.05 | 0.43 | 0.03 | 0.01 | [−0.01 to 0.03] | 0.179 |
| WBC | 8.26 | 1.97 | 8.10 | 2.04 | −0.16 | [−0.98 to 0.66] | 0.639 |
| hsCRP | 3.32 | 4.03 | 2.80 | 2.15 | −0.51 | [−2.20 to 1.17] | 0.769 |
| mMRC | 0.67 | 0.77 | 0.73 | 0.84 | 0.06 | [−0.22 to 0.34] | 0.770 |
| SGRQ activity score | 23.36 | 20.00 | 25.71 | 18.62 | 2.35 | [−5.06 to 9.76] | 0.463 |
| SGRQ impact score | 7.66 | 10.66 | 9.56 | 10.43 | 1.91 | [−2.06 to 5.88] | 0.136 |
| SGRQ symptom score | 19.54 | 18.42 | 25.79 | 23.60 | 6.25 | [−0.84 to 13.34] | 0.290 |
| SGRQ total score | 14.42 | 12.63 | 17.15 | 12.80 | 2.73 | [−1.99 to 7.45] | 0.163 |

(Continued)

| Variable | No COPD (n = 287) | | Incident COPD (n = 33) | | | | |
|---|---|---|---|---|---|---|---|
| | Mean | SD | Mean | SD | Difference | 95% CI | p[a] |
| CAT (score) | 9.46 | 6.85 | 12.07 | 7.77 | 2.61 | [−0.01 to 5.23] | 0.052 |
| MQq 1 | 1.40 | 1.35 | 1.91 | 1.55 | 0.50 | [0.00–1.01] | 0.099 |
| MQq 2 | 1.22 | 1.32 | 1.59 | 1.52 | 0.37 | [−0.12 to 0.87] | 0.207 |
| MQq 3 | 0.63 | 1.00 | 1.06 | 1.32 | 0.44 | [0.06–0.82] | 0.064 |
| MQq 4 | 0.42 | 0.54 | 0.66 | 0.55 | 0.24 | [0.04–0.43] | 0.015 |
| MQq 5 | 0.31 | 0.49 | 0.50 | 0.57 | 0.19 | [0.00–0.37] | 0.050 |
| MQq 6 | 0.63 | 0.63 | 0.75 | 0.67 | 0.12 | [−0.12 to 0.35] | 0.331 |
| MQq 7 | 0.27 | 0.58 | 0.28 | 0.58 | 0.02 | [−0.20 to 0.23] | 0.799 |
| MQq 8 | 0.25 | 0.53 | 0.25 | 0.51 | 0.00 | [−0.20 to 0.19] | 0.951 |
| MQq 9 | 0.24 | 0.51 | 0.25 | 0.44 | 0.01 | [−0.17 to 0.20] | 0.574 |
| MQq 10 | 0.56 | 0.73 | 0.69 | 0.97 | 0.13 | [−0.15 to 0.40] | 0.751 |
| MQq 11 | 1.19 | 0.91 | 1.28 | 0.89 | 0.10 | [−0.24 to 0.43] | 0.602 |
| MQq 12 | 1.30 | 0.96 | 1.38 | 0.79 | 0.07 | [−0.28 to 0.42] | 0.604 |
| MQq 13 | 0.41 | 0.73 | 0.56 | 0.80 | 0.15 | [−0.12 to 0.42] | 0.148 |
| MQq 14 | 0.33 | 0.62 | 0.31 | 0.47 | −0.02 | [−0.25 to 0.20] | 0.730 |
| MQq 15 | 0.57 | 0.73 | 0.63 | 0.71 | 0.05 | [−0.21 to 0.32] | 0.546 |
| MQq 16 | 0.41 | 0.70 | 0.41 | 0.61 | 0.00 | [−0.26 to 0.25] | 0.774 |
| MQq 17 | 1.42 | 0.78 | 1.34 | 0.65 | −0.08 | [−0.36 to 0.20] | 0.387 |
| MQq 18 | 1.32 | 0.80 | 1.38 | 0.71 | 0.05 | [−0.23 to 0.34] | 0.900 |
| MQ (total score) | 12.89 | 8.89 | 15.22 | 10.32 | 2.33 | [−1.00 to 5.65] | 0.277 |

**Notes:**
[a] Mann-Whitney test.

p/y , pack-years; BMI, body mass index; HR, heart rate; BD, bronchodilator; FVC, forced vital capacity; FEV1, forced expiratory volume in 1 s; TIFF, Tiffeneau index (FEV1/FVC); MEF25, mid-expiratory flow at 75% of FVC; MEF50, mid-expiratory flow at 50% of FVC; PEF, peak expiratory flow; DLCO, diffusing capacity of the lungs for carbon monoxide; KCO, carbon monoxide transfer coefficient; 6MWT, 6-min walk test; EBTb, baseline exhale breath temperature; EBTd, difference in exhaled breath temperature after smoking a cigarette; RBC, red blood cell count; Hgb, hemoglobin; htc, hematocrit; WBC, white blood cell count; hsCRP, high-sensitivity C-reactive protein; mMRC, modified Medical Research Council dyspnea scale; CAT, COPD Assessment Test; SGRQ, St. George's respiratory questionnaire; MQ, MARKO questionnaire; MQq, MARKO questionnaire question number.

2 years follow-up (AUC 0.79, 95% CI [0.74–0.84], $p < 0.001$) (Fig. 2). A 55% increase in odds for incident COPD was found for an increase in score to question 3, and an 243% increase in odds for increase in score to question 4 with the 11 times increased odds in male subjects.

When only active smokers were analyzed a change in EBT after a cigarette (ΔEBT) was added to a previous model (AUC 0.83, 95% CI [0.78–0.88], $p < 0.001$) (Fig. 3). A 29% of increase in odds for each 0.01 °C of EBTd, 69% increase in odds for incident COPD was found for an increase in score to question 3, and an 333% increase in odds for increase in score to question 4 with the almost 11 times increased odds in male subjects.

## DISCUSSION

Aside from smoking cessation for active smokers, there are no disease modifying therapies for COPD (Sin, 2023). By the time patients develop spirometric obstruction, they have lost nearly half of their small airways and a third of their gas-exchanging units (Koo et al., 2018)

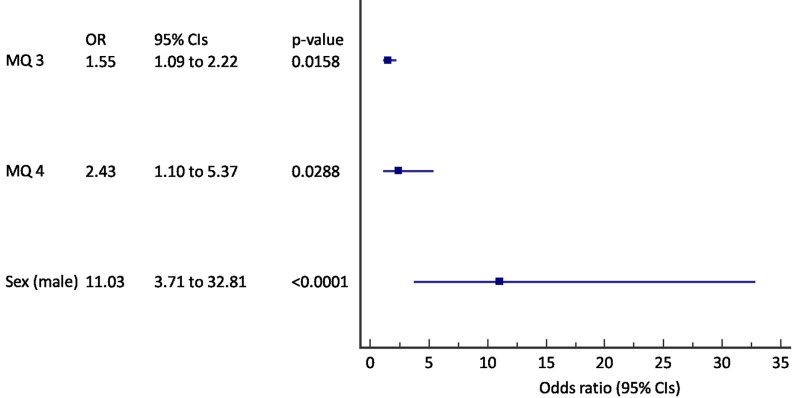

**Figure 2 Forest plot of the results of the multivariate logistic regression for the prediction of incident COPD (N = 320).** MQ, MARKO questionnaire question number; CIs, confidence intervals.

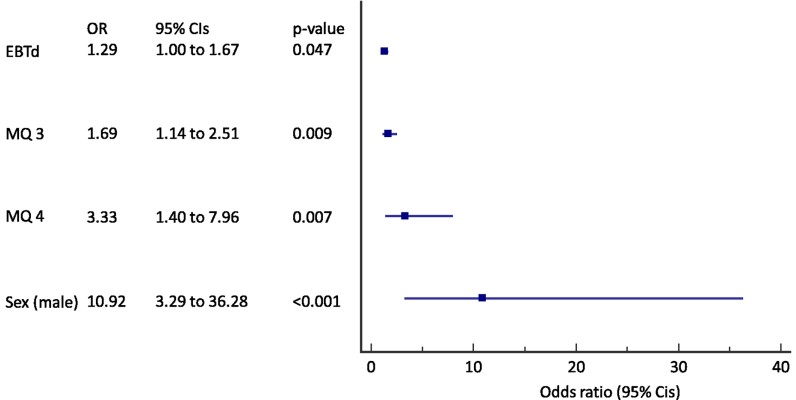

**Figure 3 Forest plot of the results of the multivariate logistic regression for the prediction of incident COPD in active smokers (N = 245).** EBTd, change in exhaled breath temperature after a smoked cigarette; MQ, MARKO questionnaire question number; CIs, confidence intervals.

and their disease is "fixed" and cannot be modified (*Stolz et al., 2022*). As lung regeneration is not possible at this point, it is crucial to identify "COPD" patients before they develop airflow limitation on spirometry and intervene at these early stages of disease. The importance of this stage is recently recognized and has different labels as "pre-COPD" and "early COPD" (*Celli et al., 2022*). No matter how we call it, for disease modification patients have to be identified at this stage of disease. More sensitive technologies (CT and hyperpolarized gas MRI, impulse oscillometry, body plethysmography and cardiopulmonary exercise tests) are required to make this diagnosis. That would implicate to test all the symptomatic smokers, but in the studies, up to 50% of smokers have some symptoms (*Woodruff et al., 2016*) but only 8% of them will develop COPD so better stratification of smokers at risk is needed.

Clinical course of COPD is often complicated by exacerbations and episodes of worsening of respiratory symptoms which contribute to disease progression. The

importance of COPD exacerbations cannot be overstated. The latest GOLD guidelines are specially addressing the problems of exacerbations and are focused on their prevention. Exacerbations can cause damage to the airways and lungs, leading to a more rapid decline in lung function over time. Addressing exacerbations promptly and effectively can improve quality of life, prevent further lung damage, and reduce healthcare costs (*Wedzicha, 2004*).

Even a single COPD exacerbation can result in a significant increase in the rate of decline in lung function (*Halpin et al., 2017*; *Dransfield et al., 2017*). Frequency of exacerbations contributes to long term decline in lung function of patients with moderate to severe COPD (*Donaldson et al., 2002*). Identification and correct assessment of COPD exacerbations is paramount, given it will strongly influence therapeutic success (*Oliveira et al., 2017*).

Exacerbations are categorized into mild, moderate, and severe ones in terms of either clinical presentation (number of symptoms) or utilization of health care resources (*Wedzicha et al., 2013*). Most of the current knowledge about COPD exacerbations is based on the evidence we have for a moderate and severe exacerbations. Mild COPD exacerbations are less well studied (*Miravitlles et al., 2004*). Even mild and unreported exacerbations might negatively affect the health-related quality of life (QOL) and lung function (*Wilkinson et al., 2004*). Based on a Japanese study, there is also the phenotype of the frequent mild COPD exacerbator but there is not enough data of the impact of this phenotype on the course of the disease (*Sato et al., 2016*).

The results of our investigation showing that the question "Have you had a severe cold with cough or bronchitis during preceding 12 months?" in the MARKO questionnaire is predictive for the development of COPD in the population at risk is moving our focus even further. The question that warrants further investigation is whether those events are the "exacerbations" of biologically present but not jet clinically overt COPD?

Exacerbations of COPD are thought to be caused by complex interactions between the host, bacteria, viruses, and environmental pollution. Respiratory infections are important triggers of acute exacerbations of COPD (*Love & Proud, 2022*). Recent findings connect the frequency and severity of LRTIs prior to COPD diagnosis with increasing rates of subsequent exacerbations and increasing risk of all-cause and COPD-related mortality (*Whittaker et al., 2022*). Little is known about the association of lower respiratory tract infections before chronic obstructive pulmonary disease and future course of the disease. Based on our results, smokers prone to frequent LRTIs could be at increased risk for the progression of lung damage to clinical COPD. Further investigation should be focused on the possibility to prevent that and cut the chain of events that lead from biological COPD to clinical one.

Other important finding from our study is that there is the association between the change in EBT after a smoked cigarette (ΔEBT) and development of clinical COPD in the patients at risk. About 8 percent of middle-aged male smokers progress to moderate COPD over five years (*Geijer et al., 2006*). Since COPD progression is associated with an enhanced chronic inflammatory response in the airways and lung tissue to harmful particles or gases (*Vestbo et al., 2013*), measuring that inflammatory response could

separate the susceptible patients from the other smokers. The inflammation observed in the lungs of COPD patients appears to be a modification of the normal inflammatory response to chronic irritants such as cigarette smoke. Inflammation in respiratory diseases causes hypervascularization and increased blood flow in the airway wall with subsequent increase in the temperature of the affected tissues, and the airway temperature can be a correlate to peripheral airway inflammation (*Tufvesson et al., 2020*).

We have previously demonstrated that there is the difference in the change of EBT after the cigarette exposure (*Labor et al., 2016a*) in patients that progress to clinical COPD. Since those patients are at greater risk for COPD development, measurement of reactivity of EBT after the acute exposure to the environmental factor (cigarette smoke or other pollutants) could be the marker of the pathophysiological COPD before the lung function decline. Strict monitoring of those patients for respiratory infections/exacerbations with early prevention of the elicit exposure could prevent the further irreversible lung injury and preserve the healthy lung function and postpone or even avert the COPD development.

There are some limitations to our study. The sample size is rather small and the expected incidence of COPD was not reached. Although we have tested new simple tools as markers of future COPD some other markers (genetic, epigenetic, metabolomics) could add some additional value. Also we didn't have a control group of non-smokers of the same age and sex. This limitations were due to budget restrictions and one should consider them when evaluating the results of our study. The positive thing is that we showed that there is at least a limited power of predictability for incident COPD that lies in simple markers like EBT or a questionnaire that could be used globally for screening. Our results however have to be corroborated and validated by other comparable studies and in a broader population.

## CONCLUSIONS

Chronic obstructive pulmonary disease (COPD) is one of the top causes of morbidity and mortality worldwide. At the time of diagnosis, there is already irreversible lung damage. To prevent that, we should define the parameters for detection of patients with early pathophysiological changes and initiate the necessary measures for preventing further lung damage. Based on our results, a simple self-administered questionnaire with questions about breathlessness and severe cold with cough or bronchitis during preceding months can detect early changes in smokers/ex-smokers. It seems that also the change of EBT after the cigarette exposure in smokers can detect early changes. Early interventions based on these results should be tested for efficacy in COPD prevention.

## ACKNOWLEDGEMENTS

The authors wish to thank the consultant Prof. Peter MA Calverley for his valuable advice regarding study design and the construction/design and review of the MARKO questionnaire together with psychologists Professors Adrijana Košćec Đuknić and Biserka Radošević-Vidaček. The authors would also like to thank the following researchers who helped in the recruitment of participants and the data collection: Vjekoslava Amerl-Šakić,

MD; Ines Balint, MD; Merim Bezdrov, MD; Mirjana Bezdrov, MD; Ivana Boban, MD; Ljiljana Bulat Kardum, MD, PhD; Ines Diminić Lisica, MD, PhD; Albina Dumić, MD, PhD; Ljiljana Ismić, MD; Tajana Jalušić Glunčić, MD; Renata Grgurić, MD; Monika Jeđud, MD; Đivo Ljubičić, MD; Tina Lukić, MD; Ljiljana Lulić-Karapetrić, MD; Suzana Maltar-Delija, MD; Dubravka Margaretić, MD; Davorka Martinković, MD; Zdenka Meštrović, MD; Nataša Mrduljaš-Đuić, MD; Jasna Nagyszombaty-Šarić, MD; Darja Nelken-Bestvina, MD; Sanja Popović Grle, MD, PhD; Sanda Pribić, MD; Jadranka Radman, MD; Rosanda Rosandić-Piasevoli, MD; Alen Stojanović, MD; Karla Tudja, MD; Neven Tudorić, MD, PhD; Bojana Vakanjac, MD; Vanja Viali, MD.

### Funding
The study was funded by an unrestricted grant from GlaxoSmithKline (GSK eTrack number: CRT114338). Prof. Davor Plavec and Dr. Žarko Vrbica as principal investigators and the Children's Hospital Srebrnjak have for the purpose of an investigator initiated study MARKO (ClinicalTrials.gov Identifier NCT01550679) received an unrestricted grant from GlaxoSmithKline. GlaxoSmithKline has not influenced the study design or study protocol, has no ownership rights over data gathered by the study and has no influence on this and further publications arising from data gathered through this study. The funders had no role in study design, data collection and analysis, decision to publish, or preparation of the manuscript.

### Grant Disclosures
The following grant information was disclosed by the authors:
GlaxoSmithKline: CRT114338.

### Competing Interests
Plavec Davor is an Academic Editor for PeerJ. Plavec Davor is employed by Prima Nova, a private health institution. The authors declare that they do not have any other competing interests.

### Author Contributions
- Žarko Vrbica conceived and designed the experiments, performed the experiments, analyzed the data, prepared figures and/or tables, authored or reviewed drafts of the article, and approved the final draft.
- Justinija Steiner performed the experiments, analyzed the data, prepared figures and/or tables, authored or reviewed drafts of the article, and approved the final draft.
- Marina Labor performed the experiments, analyzed the data, prepared figures and/or tables, authored or reviewed drafts of the article, and approved the final draft.
- Ivan Gudelj performed the experiments, analyzed the data, prepared figures and/or tables, authored or reviewed drafts of the article, and approved the final draft.

- Davor Plavec conceived and designed the experiments, performed the experiments, analyzed the data, prepared figures and/or tables, authored or reviewed drafts of the article, and approved the final draft.

## Human Ethics

The following information was supplied relating to ethical approvals (*i.e.*, approving body and any reference numbers):

The study for all investigational sites was approved by the Children's Hospital Srebrnjak Ethics Committee. The IRB approval and Consent forms were already reviewed for the already published manuscript and approved when the protocol of the study was published (http://dx.doi.org/10.1186/s12890-017-0378-6).

## Data Availability

The raw data are available in the Supplemental File.

## Supplemental Information

Supplemental information for this article can be found online at http://dx.doi.org/10.7717/peerj.16650#supplemental-information.

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
