# Peer review of "Breathlessness and “exacerbation” questions predictive for incident COPD (MARKO study): data after two years of follow-up"

_PeerJ, doi:10.7717/peerj.16650_

## Round 0.1 · original submission · Major Revisions

The reviewers have raised critical issues with your study. Please see their comments for your action.

**Language Note:** The review process has identified that the English language must be improved. PeerJ can provide language editing services - please contact us at copyediting@peerj.com for pricing (be sure to provide your manuscript number and title). Alternatively, you should make your own arrangements to improve the language quality and provide details in your response letter. – PeerJ Staff

Reviewer 1 ·

Basic reporting

The authors did not refer to the previous studies on EBT in case of COPD.

Experimental design

The study is interesting. However, the following concerns should be addressed.

Comments:
1. Studying 40-65 years of age is a broad range. Please categorize the observations for 5-10 years. So that it would be easy to further understand the effect of age factors along with smoking on COPD occurrence.

2. The control groups (no smoking) are missing in this study.

3. Please represent the significant observations of Tables in a graphical format.

4. Please provide the experimental design of this study as a schematic.

Validity of the findings

Conclusions could be improved.

Reviewer 2 ·

Basic reporting

The submitted manuscript is written in professional English but can be further linguistically upgraded to improve clarity. Still, I do not believe that any of the authors' intended meaning is lost.
The provided references appear adequate. Some preceding work of interest on EBT and smoking is missing and could shed further light on the topic (I can provide the respective references if deemed necessary).
The structure of the article complies with the standards of academic medical writing. A problem I perceive is the need to look up the MARKO questionnaire in cited articles. As the words "breathlessness" and "exacerbation" in the MARKO questionnaire constitute some of the most important findings of the study (primary end point), it would be important to find a place in the body of the text to quote those questions (alternatively, add the MARKO questionnaire as a whole in 'Supplementary material' if such an option exists).
Three tables with computed values from the analyzed variables are provided. Table 3 is rather long containing all 18 questions form the MARKO questionnaire, not listed in the manuscript: abridging this table or submitting it as "Supplementary material" would improve readability.
The graphical content of the two figures can be presented in textual format.
The results of the study substantiate the authors' research hypothesis.

Experimental design

The manuscript contains original primary research performed.
The research question is well defined addressing an important need in the early diagnosis of COPD that would have pragmatic implications.
The authors define their design as "two-phase prospective observational cohort study" (line 82).
As "two-phase" could be erroneously associated with "phase two" clinical trial, I would suggest modifying it to "prospective observational cohort study of two years duration".
The "Materials and Methods" section contains detailed information about the work done, suggesting high technical and ethical standard and allowing replication of the methods used. The statistical methods applied seem adequate for the purpose of the study.
After two years of followup the authors have identified 33 subjects with incident COPD, who are then compared to the rest of the cohort. What is missing is the description of how and by whom this identification process was done. This is of paramount importance, as the results constitute the major novelty of the study.

Validity of the findings

The essence of the study is following up a well characterized cohort of "healthy" smokers (n=320), who were reassessed after 2 years. Additionally, EBT (a surrogate marker of airway inflammation) was measured before and after smoking one cigarette, and the resulting EBT difference contributed significantly to the multiple logistic regression equation predicting development of incident COPD. This finding along with the diagnostic value of the MARKO questions on "breathlessness" and "exacerbation" constitute the most important and interesting novelties of the study. They appear to be well substantiated by the provided data and statistical analysis.
The stated conclusions are linked up to the original research done with the exception of "Respiratory infections in such a patients should be considered as exacerbations with all the consequences that they have in advanced COPD" (lines 342-343), for which the study was not designed to assess.

Additional comments

There are many minor flaws: typos (e.g. "simplex portable devices" - line 60 /instead of "simple"/; "predictivity of MARKO questionnaire" - line 77 /instead of "predictability"/), inappropriately used terms ("self-applicable ...questionnaire" - line 120 /instead of "self-administered"/) to name a few.
These would require a careful revision and correction.

---

## Round 0.2 · accepted · Accept

All reviewer's comments have been adequately addressed.

Reviewer 1 ·

Basic reporting

No comment

Experimental design

No comment

Validity of the findings

No comment

Reviewer 2 ·

Basic reporting

The authors have have complied with my suggestions and addressed my concerns satisfactorily.
I have no further comments.

Experimental design

The authors have have complied with my suggestions and addressed my concerns satisfactorily.
I have no further comments.

Validity of the findings

The authors have have complied with my suggestions and addressed my concerns satisfactorily.
I have no further comments.

Additional comments

The authors have have complied with my suggestions and addressed my concerns satisfactorily.
I have no further comments.